# Enhancing the Retail Food Environment Index (RFEI) with Neighborhood Commuting Patterns: A Hybrid Human−Environment Measure

**DOI:** 10.3390/ijerph191710798

**Published:** 2022-08-30

**Authors:** Bailey Glover, Liang Mao, Yujie Hu, Jiawen Zhang

**Affiliations:** Department of Geography, University of Florida, Gainesville, FL 32601, USA

**Keywords:** retail food environment index (RFEI), healthy food, commuting flows, transportation mode, GIS

## Abstract

The Retail Food Environment Index (RFEI) and its variants have been widely used in public health to measure people’s accessibility to healthy food. These indices are purely environmental as they only concern the geographic distribution of food retailers, but fail to include human factors, such as demographics, socio-economy, and mobility, which also shape the food environment. The exclusion of human factors limits the explanatory power of RFEIs in identifying neighborhoods of the greatest concern. In this study, we first proposed a hybrid approach to integrate human and environmental factors into the RFEI. We then demonstrated this approach by incorporating neighborhood commuting patterns into a traditional RFEI: we devised a multi-origin RFEI (MO_RFEI) that allows people to access food from both homes and workplaces, and further an enhanced RFEI (eRFEI) that allows people to access food with different transportation modes. We compared the traditional and proposed RFEIs in a case study of Florida, USA, and found that the eRFEI identified fewer and more clustered underserved populations, allowing policymakers to intervene more effectively. The eRFEI depicts more realistic human shopping behaviors and better represents the food environment. Our study enriches the literature by offering a new and generic approach for assimilating a neighborhood context into food environment measures.

## 1. Introduction

In public health, there has been a growing body of literature regarding the associations between neighborhood food environment and diet-related health outcomes, such as rates of obesity, Type II diabetes, and depression [1,2]. Meanwhile, food access has drawn increasing attention from urban/regional planners as disparities are frequently reported among racial/ethnic and socio-economic groups, and have become a significant environmental justice issue [3,4,5]. As a result, a range of indices have been developed and evolved during the last two decades to better characterize local food environments, identify where access to healthy food is limited, and help policy makers target intervention efforts [6,7].

Among these indices, the Retail Food Environment Index (RFEI), as the focus of this article, was originally proposed by the California Center for Public Health Advocacy [8]. It is defined as the number of less healthy food retailers (e.g., fast food restaurants and convenience stores) divided by the number of healthy food retailers (e.g., grocery stores and supermarkets) in an area, for example, a census tract. By comparing both types of food retailers, the RFEI is capable of indicating “food deserts”, where affordable healthy food is not accessible [9], and “food swamps”, where large amounts of energy-dense snack foods inundate healthy food options [10]. Because of its easy implementation, it has been widely adopted by government agencies and researchers in a wide range of fields. For example, the RFEI modified by the US Center for Disease Control and Prevention [11] was found to be negatively associated with body mass index among US adults [12]; Paquet et al. reported that a higher RFEI was related to increased risk of abdominal obesity in Australia [13]; Havewala used the RFEI to investigate the effects of food environment on residential segregation in US metropolitan areas [14].

The simple formulation of RFEI, however, has been criticized for its lack of consideration of human characteristics, such as population structure, socio-economic status, and mobility patterns, which also shape the neighborhood food environment [15,16]. The RFEI is purely environmental as it only accounts for the spatial distribution of food retailers. Consequently, when the RFEI (and its variants) was analyzed against people’s dietary behavior or outcomes, the findings were not consistent in the literature. For instance, some existing work shows a statistically significant association between RFEI and obesity [1,17,18], but some other research indicates no such association [19,20,21]. Several studies have found that RFEI is associated with people’s healthy food intakes [22,23], while a few have indicated null associations [20]. No matter how the RFEI has been modified in the last decade, changes have been predominately made regarding food retailers, but little attention has been paid to human factors.

The aim of this study is two-fold. First, we propose a generic approach to integrate human factors into the current framework of RFEI. This hybrid human−environment approach can be applied to integrating any human factors, such as the population structure and socio-economic status, and mobility. Second, we demonstrate this approach with a specific example of incorporating people’s commuting patterns to measure the food environment of Florida, USA.

## 2. Review of Food Environment Literature: Factors, Measures, and RFEIs

Pioneering work by Glanz et al. [24] proposed a two-level concept to describe food environments: the “community-level environment”, which consists of the number, type, location, and accessibility of food retailers/stores, and the “store-level environment”, which include the availability, quality, price, and variety of food within stores. However, this concept is mainly focused on the physical environment, but lacks concern for human factors that may also shape people’s shopping behavior, such as spatial mobility, social segregation, and purchasing power. Turner et al. [16] refined the concept of the food environment by adding human characteristics and proposed a two-domain framework: the “external domain”, which refers to the presence of food products (availability), prices, quality, and other vendor properties, and the “personal domain”, which concerns individuals’ travel patterns over space (accessibility), purchasing power, time budget, and food preferences. From a methodological perspective, the external domain is typically measured by a static GIS-based approach that maps vendors’ count number, ratio, or distance-based proximity in a given neighborhood, such as a census tract [16,25]. These measures are also referred to as “place-based” measures, for example, the food access indicator used by the U.S. Department of Agriculture (USDA) [26] and the RFEI as the focus of this paper. The personal domain can be measured by a dynamic activity-based approach that tracks individuals’ daily and purchasing activities using travel surveys, qualitative questionnaire, and GPS devices [15,16,27,28]. These measures are thus referred to as “people-based” measures, which emphasize human demographics, travel, and shopping behavior.

As both domains interact with one another to influence the food environment, there has been a new trend in recent years to devise hybrid measures that cross two domains. For example, under a supply−demand framework, gravity-based models [6] and floating catchment area models [29,30] have been adopted to consolidate food retailers, population demands, and travel time into hybrid measures of food access for neighborhoods. Shannon’s study [15] suggested using mixed research methods to examine how store characteristics, neighborhood context, and individual mobility interact to shape food provisioning practices. Despite the popularity of REFI, its evolution in the last decade has been limited in the external domain, and few studies have attempted to expand it to the personal domain. In particular, existing RFEIs do not represent human mobility sufficiently. These indices simply assume that all people travel from their homes as a single type of origin, while ignoring trips made to food retailers that are beyond their residence, for example from people’s workplaces or on their way home [31,32]. Furthermore, current REFIs fail to differentiate people’s travel modes to food retailers [33], but in reality people can shop using various travel modes, such as by car, public transit, bicycle, and walking [34]. These simple assumptions about human mobility could largely limit the explanatory power of RFEIs in identifying neighborhoods of the greatest concern.

## 3. Methodologies

### 3.1. The Generic Approach

To address the limitations of RFEIs, we propose a hybrid approach to incorporate human factors. Generally, this approach first stratifies a studied population by any characteristics of interest (e.g., age groups, income levels, and mobility patterns), then calculates the RFEI for each subpopulation, and finally aggregates them as a weighted sum.

To demonstrate how to apply this approach, we considered people’s commuting pattern (where and how they go to work) as a specific example, because it significantly affects the spatial extent to which people can access healthy food, the so called “food catchment area” [6]. Details of implementation and a case study in Florida, USA, are described as follows.

### 3.2. mRFEI as a Single-Origin Index

We utilized the CDC’s mRFEI [11] as a basis for enhancement and comparison, as it is used throughout the US. The index assumes that a study area is composed of *N* neighborhoods, such as census tracts, with each having a population denoted as *Pi* (=1, 2, 3…, *N*). All food retailers in the study area are classified into either healthy or less healthy categories. People travel to food retailers only from their home neighborhoods as the single origin. The resulting food catchment area incorporates the home neighborhood itself and a Euclidean buffer of 0.5 miles from the neighborhood’s boundary (Figure 1A). The mRFEI of neighborhood *i* is expressed as a ratio between the number of healthy food retailers and all food retailers in the neighborhood’s catchment area (*CAi*), formulated as follows:(1)$$mRFEI_{i}=\frac{\sum_{k}hr\left(i,k\right)}{\sum_{k}hr\left(i,k\right)+\sum_{k}lhr\left(i,k\right)}\times 100\;\;\;\;\;\;\;$$
(2)$$hr\left(i,k\right)=\{\begin{array}{c} 1,\;\;If\;a\;healthy\;food\;retailer\;k\;is\;within\;CA_{i}\\ 0,\;\;\;Otherwise \end{array}$$
(3)$$lhr\left(i,k\right)=\{\begin{array}{c} 1,\;\;If\;a\;less\;healthy\;food\;retailer\;k\;is\;within\;CA_{i}\\ 0,\;\;\;Otherwise \end{array}$$
where  hr(i,k) and lhr(i,k) are discriminant functions to identify if a healthy or less healthy food retailer *k* (=1, 2, 3…, K) falls within the catchment area of neighborhood *i*. A larger mRFEI value indicates a heathier food environment.

### 3.3. Multiple Origin RFEI (MO_RFEI)

Based on the mRFEI, we added people’s work neighborhoods as another origin for food shopping, and thus created a multiple-origin RFEI (MO_RFEI). Here, a study area is composed of *N* neighborhoods of a population *P_i_* (=1, 2, 3…, *N*). The population residing in neighborhood *i* can commute to work in any neighborhood, including itself (Figure 1B). Hence, any neighborhood can be a home neighborhood for residence, but also a workplace neighborhood to which people commute. We divided the population of neighborhood *i* (*P_i_*) into subpopulations *P_ij_*, which denotes the number of people living in neighborhood *i* and traveling to neighborhood *j* for work. As illustrated in Figure 1B, both neighborhoods *P*_1_ and *P*_2_ have certain commuters traveling to a common neighborhood *P_3_* for work, labeled as *P_13_* and *P_23_*. For implementation, we still adopted the 0.5-mile buffer to delineate food catchment area. Therefore, commuters *P_13_* could access food retailers, not only within a 0.5-mile buffer around its home neighborhood, but also food retailers in a 0.5-mile buffer around its work neighborhood, and so does *P_23_*. There can be overlap between food catchment areas between neighborhoods *P*_1_ and *P*_2_.

For each subpopulation *P_ij_*, we calculate the mRFEI(i,j) as a ratio between the number of healthy food retailers and the total number of food retailers that are considered accessible by *P_ij_*, formulated as follows:(4)$$mRFEI\left(i,j\right)=\frac{\sum_{k}hr\left(i,k\right)+\sum_{k}hr\left(j,k\right)}{\sum_{k}hr\left(i,k\right)+\sum_{k}hr\left(j,k\right)+\sum_{k}lhr\left(i,k\right)+\sum_{k}lhr\left(j,k\right)}$$
where *hr(i,k)* and *lhr(i,k)* can be calculated based on Equations (2) and (3). Then, the MO_RFEI of entire neighborhood *i* is computed as a population weighted sum of the mRFEI(i,j) across all work neighborhoods, formulated as Equation (5). A lower MO_RFEI indicates that most people living in this neighborhood have worse access to healthy food at both the home and workplace.
(5)$$MO\_RFEI_{i}=\frac{\sum_{j=1}^{N}mRFEI\left(i,j\right)\cdot P_{ij}}{P_{i}}\times 100\;\;\;$$

### 3.4. Enhanced RFEI for Multiple Origins and Multiple Modes (eRFEI)

From the MO_RFEI, we further added multiple travel modes of commuting to formulate an Enhanced RFEI (eRFEI). Instead of using a 0.5-mile buffer to define the food catchment area, we related the delineation to different travel modes. That is, the food catchment area is the maximum spatial extent that can be reached within a time limit and by a specific travel mode (speed). Here, we further divided a subpopulation *P_ij_*, who commute between neighborhood *i* and *j,* by transportation mode and denote *P_ijm_* as the commuting population between neighborhood *i* and *j* with a transportation mode of type *m* (=1, 2, 3, …, M). As travel modes are considered, we used a threshold travel time *t_0_* to delineate distinct catchment areas for various modes. As shown in Figure 1C, commuters between neighborhoods *P*_1_ and *P_3_* were further split into *P_13m_* by *m* = car, public transit, bike, or walking. Each *P_13m_* has food catchment areas at both the home and workplace that are demarcated by the travel time limit *t_0_* and mode *m.* Different from the previous two indices, the food catchment areas can vary within a neighborhood, as some people can travel further than others due to mode choices, and thus have access to more food retailers.

For each subpopulation *P_ijm_*, we calculate *mRFEI(i,j,m)* as the ratio between the number of healthy food retailers and the total food retailers accessible from a home neighborhood *i* and a work neighborhood *j* via a mode *m*:(6)$$\begin{array}{l} mRFEI\left(i,j,m\right)\\ =\frac{\sum_{k}hr\left(i,k,m\right)+\sum_{k}hr\left(j,k,m\right)}{\sum_{k}hr\left(i,k,m\right)+\sum_{k}hr\left(j,k,m\right)+\sum_{k}lhr\left(i,k,m\right)+\sum_{k}lhr\left(j,k,m\right)} \end{array}$$
(7)$$hr\left(i,\;k,m\right)=\{\begin{array}{c} 1,\;\;\;t_{i,k,m}\leq t_{0}\\ 0,\;\;\;Otherwise \end{array}$$
(8)$$lhr\left(i,\;k,m\right)=\{\begin{array}{c} 1,\;\;\;t_{i,k,m}\leq t_{0}\\ 0,\;\;\;Otherwise \end{array}$$
where $t_{i,k,m}$ is the travel time from either a home or work neighborhood *i* to a food retailer *k* by mode *m*. hr(i,k,m) and lhr(i,k,m) are discriminant functions to determine if a healthy or less healthy food retailor *k* can be reached within a threshold time $t_{0}$. Finally, we weighed each *mRFEI(i,j,m)* by the size of subpopulation and summed them by work neighborhood and by travel mode into the *eRFEI* for neighborhood *i*, formulated as follows:(9)$$eRFEI_{i}=\frac{\sum_{m=1}^{M}\sum_{j=1}^{N}mRFEI\left(i,j,m\right)\cdot P_{ijm}}{P_{i}}\times 100\;\;$$

In the following sections, we illustrated the implementation of three food environment indices, i.e., the mRFEI, MO_RFEI, and eRFEI, for Florida census tracts, and analyzed how the incorporation of multiple origins and multiple travel modes might alter the existing characterizations of food environments.

## 4. Case Study: Re-Evaluating Food Environments in Florida

### 4.1. Study Area

According to the mRFEI used by the US CDC, a great number of census tracts in Florida had a score under the national average at 10. We suspected that this measure could overestimate the number of underserved census tracts, because it failed to account for people shopping from their workplaces and ignored the effects of different travel modes. In this case study, we used MO_REFI and eREFI to re-estimate Florida’s food environment by adding multiple origins (homes and workplaces) and various travel modes (car, bus, bike, and walking).

### 4.2. Data Collection

To parameterize the three food environment indices mentioned above, the following datasets were collected for neighborhoods, commuting patterns, food retailers, and transportation networks.

#### 4.2.1. Census Tracts as Neighborhoods

We retrieved the geographic boundaries and total populations of 4172 census tracts from the US Census Bureau [35] as a representation of home and work neighborhoods. For each census tract, the geographic boundary (as polygon) was converted to a population weighted mean center (as point) based on finer census block populations within the census tract (Figure 2A). We assumed that people residing or working in a census tract travelled from the population mean center to food retailers. The total population size of each census tract was used in Equations (5) and (9) for the purpose of the weighted sum.

#### 4.2.2. Commuting between Neighborhoods

We acquired the US Census Transportation Planning Products (CTPP) data [36] to depict people’s commuting behavior between home and work neighborhoods (Figure 2A). The CTPP data record three parts of information: (1) the number of resident workers in each tract, (2) the number of employment opportunities in each tract, and (3) the number of workers aged 16 and older commuting between any pair of tracts, as shown in Figure 2A. The commuting flow data in (3) were further organized by modes of travel, including car, bus, bicycle, walking, and others. For each travel mode, we constructed a 4172 by 4172 matrix to record the commuting flow between any possible pair of home and work neighborhoods. For the non-working population (under 16 years and over 65 years), we assumed they lived and “worked” in the same tract, and their choices of travel modes followed the same distribution as the working population.

#### 4.2.3. Food Retailers

Healthy and less healthy food retailers were defined by the US CDC [11] according to the North American Industry Classification System (NAICS) of 2011. We slightly modified these definitions to accommodate the latest NAICS published in 2017 (Table A1 in Appendix A). We then retrieved a total of 4380 healthy and 17,921 less healthy food retailers in Florida (Figure 2B) from a business database [37], which provides detailed information concerning business characteristics and geographic locations.

#### 4.2.4. Transportation Networks and Travel Time Estimation

To estimate the travel time between neighborhoods and food retailers, the road network and public transit network were retrieved and compiled (Figure 2C). The road network, as a US Census TIGER/Line product [35], was utilized to estimate travel time by car, bike, and walking. In particular, travel time by car was estimated using a road speed setting described in Table A3 in Appendix A. Travel time by bike and walking was calculated with constant speeds of 16.7 km/h (10 miles per hour) and 4.8 km/h (3 miles per hour), respectively [38]. To estimate travel time by bus, we retrieved General Transit Feed Specification (GTFS) data from the Florida Department of Transportation [39] to build 31 regional transit networks. The GTFS data contain information about bus schedules, stops, and routes. ArcGIS was used to import the road and GTFS data to create networks. Then, the Network Analyst tool was used to estimate the travel time matrix for each travel mode for all possible pairs of neighborhoods and food retailers.

The census tracts and food retailer data were used to estimate the mRFEI for each census tract in Florida using Equations (1)–(3). After adding the CTPP data regarding the commuting patterns of neighborhoods, we calculated the MO_RFEI with Equations (4) and (5). By further incorporating different transportation modes and networks, we used Equations (6)–(9) to assess the eRFEI. Here, we chose a travel time threshold of 20 min (*t*_0_) to define a catchment area. This value was determined by calculating the median travel time of trips made for “purchasing goods” in the National Household Travel Survey [40]. We compared these three indices statistically and spatially to examine how the added features might alter the existing characterizations of food environments. A map illustrating a statewide comparison of these indices is offered in Figure 3.

## 5. Results and Discussion

### 5.1. Statistical Comparison

Figure 4 compares the centrality and variability of the three indices across Florida. From the mRFEI, to MO_RFEI, and to eRFEI, the results indicate a slow decrease in the mean value, but a dramatic drop in the standard deviation.

This decreasing trend can be explained by the expansion of food catchment areas when multiple origins and multiple modes were gradually incorporated. As illustrated in Figure 1A, there is no overlap between the catchment areas of two populations (*P*_1_ and *P*_2_). The MO_RFEI expands the food catchment areas by incorporating 0.5 mile buffers around workplace tracts, and thus *P*_1_ and *P*_2_ can have certain overlap in their food catchment areas (Figure 1B). Finally, by considering multiple travel modes, the eRFEI allows those who use cars and public transit to travel further, thus allowing both populations to have even larger food catchment areas and more overlap between them (Figure 1C). As food catchment areas expand from the mRFEI, to MO_RFEI, and eRFEI, the total number of accessible food retailers (the denominator in Equations (4) and (6) increases faster than the number of accessible healthy food retailers (the numerator in Equations (4) and (6), thus resulting in a decline in index scores. Meanwhile, as the catchment areas expanded, they overlapped more between neighborhoods, making the food environments more similar to each other and hence less standard deviation in the index scores.

Locating underserved neighborhoods is one of the most important usages of food environment indices. Relevant to this study, we selected ten as the cut off value for “underserved” food environments or neighborhoods, which was the national average of mRFEI estimated by the CDC [11]. We also used a value range between 10 and 20 to define “slightly underserved” food environments. Census tracts with a score above 20 were considered “well served” (Figure 3). As shown in Table 1, the mRFEI identified the greatest number of underserved neighborhoods, followed by the MO_RFEI, while the eRFEI identified the fewest. This reduction represents a change in underserved population of nearly 3.09 million people, or 15.7% of the total population of Florida. In the slightly underserved category, the results followed an opposite trend, in which the mRFEI indicated the fewest number of tracts, followed by the MO_RFEI, while the eRFEI indicated the most. Many census tracts that were classified as “underserved” by the mRFEI and MO_RFEI were moved into the “slightly underserved” category when transitioning to the eRFEI. The observed trends in both categories can be attributed again to the incorporation of multiple origins and multiple travel modes, which pulled the food environment scores toward the centrality (around 20), as discussed above. These results highlight how commuting data continue to be paramount in replicating real world travel behaviors.

### 5.2. Urban and Rural Disparities

Urban−rural disparities in access to healthy food is a major concern in public health research. Here, we used the US Census definition [41] to determine urban and rural areas: urban areas are census tracts falling within continuously built-up areas with populations of 50,000 people or more, whereas census tracts outside of the urban areas are considered rural. Our results (Table 2) are consistent with similar studies that highlighted rural residents as being at a disadvantage when accessing healthy foods compared with their urban peers [30]. The means of the food environment scores were higher among urban areas across all three indices, while the proportions of underserved populations were larger in the rural areas.

Second, our results imply that the disparities between urban and rural areas gradually narrowed: the disparity was 12.64% in the mRFEI, 12.48% in the MO_RFEI, and 4.51% in the eRFEI. As discussed above, the inclusion of multiple origins and travel modes allowed for more overlap in food catchment areas between urban and rural populations. As their food environments became more and more similar, the disparity between urban and rural areas was reduced.

### 5.3. Underserved Populations by Travel Modes

In many previous studies, the food index was calculated separately for each travel mode by forcing the entire population to choose only one travel mode [6,42]. Subpopulations by travel model were not often considered. The assumption of a homogeneous population cannot stand in many neighborhoods where people’s travel mode choices are far more diverse. In addition, this one-population-one-mode analysis prevents researchers from examining subpopulations in a neighborhood, i.e., how many people are underserved by each travel mode in a neighborhood, which is a key to precisely targeting intervention.

Superior to the other indices, the eRFEI allows for an estimation of underserved populations by travel mode. Figure 5 reveals a discrepancy of underserved populations across different travel modes in Florida. Firstly, the underserved population who commute by car is about 3.5 times larger in rural areas when compared with their urban counterparts. This result implies that far more rural car commuters need to travel more than 20 min to access healthy food when compared with urban car commuters. This can be attributed to the limited number of food retailers in rural areas, let alone that they are scattered across a wide geographic extent. Next, in urban areas, the number of underserved people commuting by walking or bus is more than 30 times higher than that of rural areas. This can be explained by the heavy reliance on walking and public transport in urban areas, and low walkability to public transport among rural census tracts.

### 5.4. Policy Implications

Several policy implications can be derived from our results. First, the traditional RFEI framework (mRFEI) may routinely exaggerate underserved areas and populations for its over-simplification of people’s travel behavior or, more generally, human factors. Our analysis highlighted a significant reduction of underserved census tracts from 806 identified by the mRFEI to only 619 by the MO_RFEI, and 55 by the eRFEI. By utilizing the more realistic MO_RFEI or eRFEI, state governments can concentrate limited resources and manpower to fewer areas that need help the most. For instance, if people in a neighborhood are underserved at home but well served at the workplace, the traditional mRFEI cannot tell the difference because it only concerns the home neighborhood. It is likely to score a low value and indicate a high priority of intervention. However, the MO_RFEI that considers both home and work would have a relatively higher score and recommend a lower priority of intervention. In a case that the MO_RFEI is low for a neighborhood, it implies that most people living in this neighborhood have worse access to healthy food at both home and the workplace. Therefore, policymakers can directly target intervention to the home neighborhood to improve the overall accessibility.

Second, we found, from the eRFEI (Figure 3), that more than half of underserved tracts were highly rural areas, clustered in the panhandle, central, and southwestern regions of Florida, where a large size of the population commuting by car were underserved (Figure 5). These areas could benefit from deploying a “produce on the go” strategy, in which a mobile farmers’ market conducts weekly trips to targeted areas and provides healthy food for purchase [43]. Although this strategy is subject to the high costs of long-distance travel, the relatively small number of underserved census tracts in Florida, as well as their spatially clustered distribution, make the strategy easier to be operationalized in the state. For example, the cities of Niceville, Orlando, and Naples can be set as distribution hubs for the panhandle, central, and southwestern regions, respectively. These cities have approximately the same distance to underserved rural census tracts in their respective regions, thus minimizing the costs for storing, loading, and delivering produce on a weekly basis.

Besides rural areas, it is worth noting that some urban areas also deal with the prevalence of poor food environments (Table 2). Figure 6 highlights 23 underserved census tracts from five urban areas. It turned out that these urban census tracts held a high percentage of commuters who relied mainly on walking and bus public transit. For these urban census tracts, a strategy of weekend farmers’ markets has been proven effective [44]. That is, by placing farmers’ markets close to public transit hubs or places with high walkability, policy makers could offer an accessible option for those urban residents to purchase healthy food.

### 5.5. Generalizability and Limitations

Although the case study focuses on commuting patterns, the proposed approach can be adapted to incorporate any other human factors into the RFEI, such as racial/ethnic composition, income, and education levels. More generically, researchers can start with stratifying a neighborhood population into subgroups by any one or more characteristics. Then, the RFEI can be calculated for each subgroup and finally aggregated by weighted sum. A key of adaptation is to determine the food catchment area for each subgroup. This needs support from travel surveys or GPS tracking technology to sample each subgroup and to specifically estimate how far they are willing to travel for food. Secondly, as all datasets used in this study are publicly available and cover the entire United States, our proposed indices can be easily expanded to other states. We also shared our Python codes and sample datasets for researchers in order to replicate them in other states. We advocate the CDC or every state government to consider our proposed index, re-estimate food environments, and adjust their interventions accordingly.

Our proposed indices were subject to several limitations. First, this study considered only two origins to approximate real-world food shopping behaviors; however, the inclusion of more trip origins for purchasing food (e.g., along commuting routes) may improve the food environment index further. Evidence of this improvement can be seen in the use of GPS in the existing literature, where several recorded stops may provide additional origins for food shopping [28]. Our proposed MO_RFEI and eRFEI can be modified to accommodate three or more origins. Secondly, because the CTPP data only contains information regarding working populations, we simply assumed that non-working populations followed the same modal distribution for food shopping. More information regarding the shopping behavior of the elderly people (65 years and over) would greatly improve the accuracy of the indices. Thirdly, the 20 min threshold used in the eRFEI calculation was derived from a Florida household travel survey that focused primarily on urban areas. The time threshold in rural areas was set to be the same as urban areas, but could differ by residence [26]. An investigation of food shopping behaviors in rural areas is needed in order to develop more suitable travel thresholds. Lastly, there may be food retailers not included in the business database. This could be especially relevant in rural communities where small food stands and weekend farmers’ markets may serve as a primary means for food attainment. All of these limitations do not diminish the value of our hybrid human−environment framework, but suggest more efforts toward data collection and model sophistication.

## 6. Conclusions

We propose a hybrid human−environment approach to enhance the RFEI, which is widely used to measure the local food environment. This approach measures the food environment from both spatial and human dimensions. Therefore, the results can be studied from either dimension, respectively, i.e., by area unit (in Figure 3) or by population characteristics (in Figure 5). Furthermore, the proposed measures allow policymakers to explore both dimensions jointly, for example, pinpointing which subpopulation in which area for intervention (e.g., Figure 6). The introduction of subpopulations into the RFEI framework allows policy makers to closely examine and design policies for a specific group of people, not necessarily for the entire population in a neighborhood, thus achieving better cost-effectiveness. We believe this hybrid approach gives an added value to the food desert/swamp literature and could provide policymakers more flexibility to tailor intervention strategies. Another contribution is to free people’s food shopping behavior from only one origin and one travel mode, which is a common assumption of current RFEIs. The enhanced RFEI offers a more realistic depiction of people’s shopping behaviors by incorporating multiple origins and multiple modes, hence providing policymakers a better estimation of the food environment.

## Figures and Tables

**Figure 1 ijerph-19-10798-f001:**
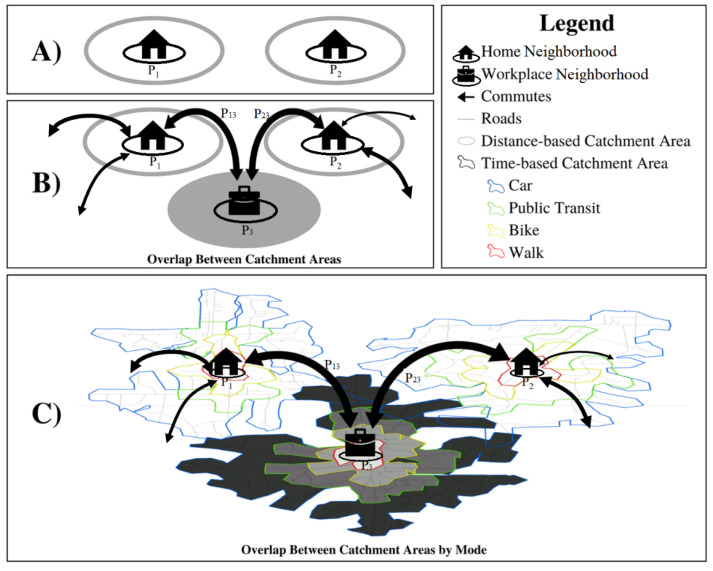
An illustration and comparison of food catchment areas defined by (**A**) the traditional mRFEI assuming people access food only from homes, (**B**) MO_RFEI with people accessing food from both home and workplace due to commuting, and (**C**) eRFEI with people accessing food from multiple origins and travel modes. Shaded areas show where the catchment areas of two populations overlap. Colored borders delineate catchment areas of different travel models.

**Figure 2 ijerph-19-10798-f002:**
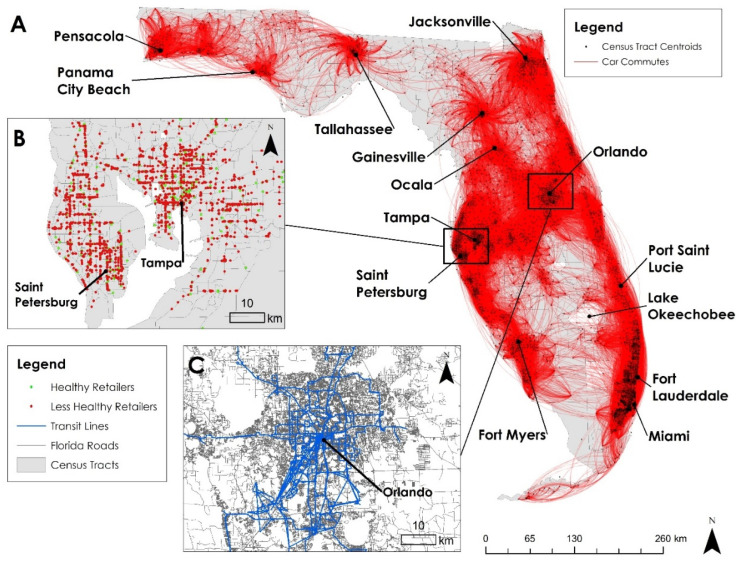
(**A**) Commuting flows by car between census tracts derived from the CTPP data. The line width is proportional to the number of commuters. For clarity, commuting trips were filtered if longer than 161 km (100 miles). (**B**) Healthy and less healthy food retailer locations in the Tampa Bay area. (**C**) Road network and transit lines in Orlando.

**Figure 3 ijerph-19-10798-f003:**
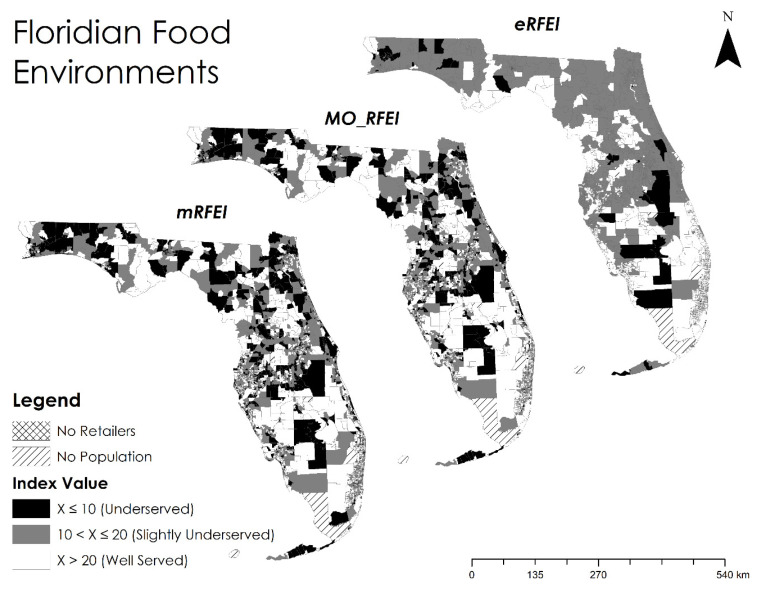
The mRFEI, MO_RFEI, and eRFEI by census tract in Florida.

**Figure 4 ijerph-19-10798-f004:**
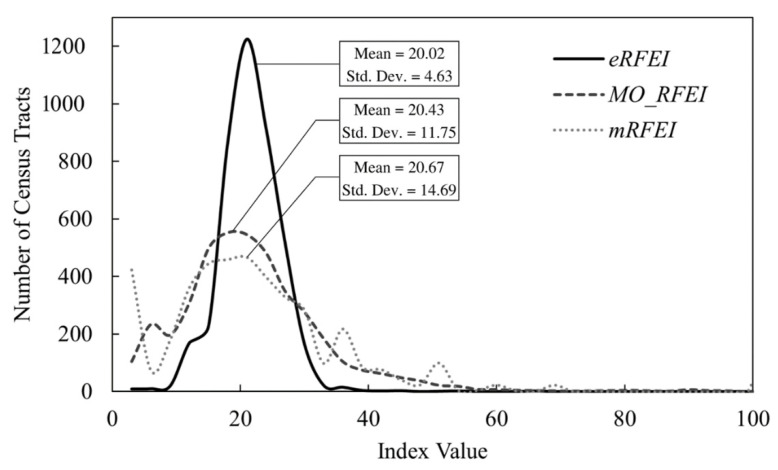
Frequency distributions of the three food environment indices with their means and standard deviation.

**Figure 5 ijerph-19-10798-f005:**
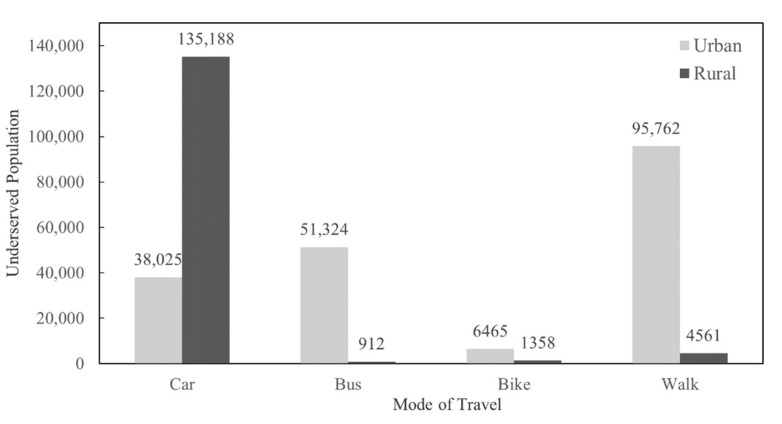
Comparison of underserved populations (estimated from the eRFEI) by residence and travel mode.

**Figure 6 ijerph-19-10798-f006:**
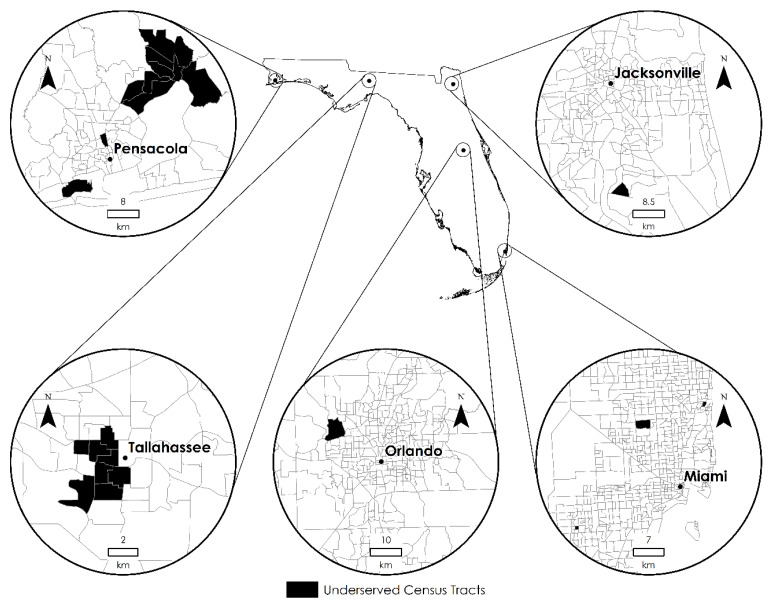
Underserved census tracts in urban areas according to the eRFEI. There are ten census tracts in Pensacola, eight in Tallahassee, one in Jacksonville, one in Orlando, and three Miami.

**Table 1 ijerph-19-10798-t001:** Statistical comparison of the three food environment indices in Florida.

Index	≤10 (Underserved)		10 < X ≤ 20 (Slightly Underserved)	
	# of Tracts	Population	# of Tracts	Population
mRFEI	806	3,355,703	1539	7,628,399
MO_RFEI	619	2,445,514	1651	8,210,267
eRFEI	55	264,499	2158	10,226,600

**Table 2 ijerph-19-10798-t002:** Urban and rural comparison of Floridian food environments.

Index	Urban			Rural		
	Mean	Underserved Tracts	Underserved Population *	Mean	Underserved Tracts	Underserved Population **
mRFEI	20.08	588	15.17%	19.89	2168	27.81%
MO_RFEI	20.58	427	10.56%	19.64	192	23.05%
eRFEI	20.29	24	0.66%	18.57	31	5.17%

*: Calculated using the total urban population in Florida as 16,872,261 in 2015. **: Calculated using the total rural population in Florida as 2,996,088 in 2015.

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
