# Peer review of "Enhancing the Retail Food Environment Index (RFEI) with Neighborhood Commuting Patterns: A Hybrid Human−Environment Measure"

_ijerph, 2022, doi:10.3390/ijerph191710798_

Round 1

Reviewer 1 Report

This is an interesting study that will make a nice addition to the food environment literature. The issue of whether people travel (and how far) to access food tends to be a limitation of community food environment studies. My main concerns are 1. the arguments about the need for the new indexes are not very clear in the introduction/literature review section and 2.the methodology is not sufficiently clear to be replicated, I really struggled to understand how the indexes were calculated. Some specific comments are below:

- Introduction: Please spell out 'demand' - ¿do you mean population density? 

- Line 77 - it's density not proximity right?

- Line 104 - what do you mean by a 'generic enhancement approach'. Please clarify

- Is the MO_RFEI a characteristic of food access of a particular individual rather than an area-based measure?  Perhaps I am not understanding this correctly. The way it is described is not clear enough to replicate. 

-If MO_RFEI is combining information on different neighborhoods which are not necessarily next to each other, how is this useful from a policy perspective? (i.e. a policy maker will not know where to intervene)

- Does each residential census tract have a single 'common workplace tract'? If this is the case, could you add some descriptives to justify this assumption (that most people in a track travel in the same direction/to the same place)?

Case study

- You say findings of RFEI could be biased in Florida. do you think food deserts are over or under estimated?  Please clarify in text.

- What do you mean by 'we retrieved xx census tracts from US CEnsus Bureau'? What information did you retrieve for each census tract?  Please clarify in text.

- It is not clear how you calculate travel time (page 6 line 181). Where did food retail data come from? What does population have to do with travel time? Please clarify in text. This is better describe below (line 206), it is redundant here (line 181).

- Can you add details of the size of 'food catchment areas' for each of the indices. 

- The results for underserved populations by travel modes again suggest that the rfei is being a applied to population groups and not areas. These results are hard to interpret and link to the food desert/swamp literature.

- The discussion clarifies many of the questions I had above. I strongly suggest you revise the early sections of your manuscript to ensure that the arguments for your sophisticated indexes are more clear. The first section of the paper (before case study) would benefit greatly from english language editing.

Author Response

Dear the editor and reviewer,

We appreciate your time and feedback on our work. We think the manuscript has been significantly improved by the revisions we made in response to your suggestions. Please find our response to each comment one by one in the attached file. Our edits in the manuscript are marked using the 'Track changes' function.

Liang Mao

Reviewer 2 Report

The topic of the article is interesting but in my opinion some implementations are necessary. The bibliography must be strengthened on the use of the indicator The retail food environment is connected for example to the body mass index. The methodology is clear. The main results of the application model proposed by you should be discussed with the reference literature

Author Response

(The authors gave the same response as above.)

Round 2

Reviewer 1 Report

I thank the authors for their thorough response to my comments. They have clarified all my questions. This is a very nice paper. I just recommend the journal/authors to give the paper a final english language edit because there are a few gramatical errors.   

Reviewer 2 Report

In my opinion, the article could be published